# N, P Self-Doped Porous Carbon Material Derived from Lotus Pollen for Highly Efficient Ethanol–Water Mixtures Photocatalytic Hydrogen Production

**DOI:** 10.3390/nano12101744

**Published:** 2022-05-20

**Authors:** Jing-Wen Zhou, Xia Jiang, Yan-Xin Chen, Shi-Wei Lin, Can-Zhong Lu

**Affiliations:** 1College of Chemistry and Materials Science, Fujian Normal University, Fuzhou 350007, China; xmzhoujingwen@fjirsm.ac.cn; 2CAS Key Laboratory of Design and Assembly of Functional Nanostructures, Fujian Provincial Key Laboratory of Nanomaterials, Fujian Institute of Research on the Structure of Matter, Chinese Academy of Sciences, Fuzhou 350002, China; xmjiangxia@fjirsm.ac.cn (X.J.); xmlinshiwei@fjirsm.ac.cn (S.-W.L.); 3Xiamen Institute of Rare-Earth Materials, Haixi Institutes, Chinese Academy of Sciences, Xiamen 361021, China; 4School of Chemistry and Chemical Engineering, Jiangxi University of Science and Technology, Ganzhou 341000, China

**Keywords:** biomass-carbon, hierarchical porous structure, photocatalytic, hydrogen production

## Abstract

Porous biochar materials prepared with biomass as a precursor have received widespread attention. In this work, lotus pollen (LP) was used as the carbon source, a variety of the pollen carbon photocatalyst were prepared by a two-step roasting method. A series of characterizations were carried out on the prepared samples, and it was found that the average particle size was about 40 μm. They also exhibit a typical amorphous carbon structure and a porous structure with a network-like interconnected surface. The photocatalytic hydrogen production performances of lotus pollen carbon (LP-C) and commercial carbon black (CB) were measured under the full spectrum illumination. LP-C-600 showed the best hydrogen production performance (3.5 μmol·g^−1^·h^−1^). In addition, the photoelectrochemical (PEC) measurement results confirmed that the LP-C materials show better incident photon-current efficiency (IPCE) performance than the CB materials in the neutral electrolyte. This is because the unique surface wrinkling, hierarchical porous structure, and the N, P self-doping behavior of the LP-C samples are able to improve the light utilization efficiency and the carrier separation/transfer efficiency, thereby further improving the overall hydrogen production efficiency.

## 1. Introduction

The energy problem has always been a worldwide issue that has attracted much attention. The current energy structure is still mainly composed of fossil energy such as coal, natural gas, and oil, but these fossil energy sources have shown irreversible harm to the environment [1]. In this case, there is an urgent need for researchers to develop renewable green energy sources [2,3]. However, hydrogen energy has received more and more attention due to its clean, easy-to-obtain and low-cost advantages [4]. At present, the main hydrogen production methods mainly include thermochemistry, electrolysis, and photocatalysis [5,6]. Thermochemical method is one of the main hydrogen production routes used in modern society, by which the hydrogen was produced by cracking petroleum or reforming natural gas with steam methane [7]. However, carbon dioxide, as an environmentally polluting gas, is inevitably produced in the process of thermochemical hydrogen production, meaning this method is not sustainable [8]. High-purity hydrogen can also be produced by the water electrolysis reaction. However, this process not only faces the problem of high energy consumption, but can also be environmentally unfriendly if we consider that the electricity may come from the burning of carbon-containing fuel [9,10,11]. Currently, the photocatalytic process is the most environmentally friendly and sustainable technology for hydrogen production [12,13], as it is mainly a method for producing hydrogen without pollution gas, driven by inexhaustible solar energy [14].

Biomass-derived carbon materials have received more and more attention due to the diversity of their microstructures [15,16] and composition [17,18]. However, it is currently mainly used for photocatalytic degradation of pollutants [19,20,21], and there are few studies on photocatalytic hydrogen production. At the same time, N doping is often used to improve photocatalytic performance [22]. However, the application of P, which is commonly used for the treatment of catalyst in industry, is less involved in photocatalysis [23,24]. In fact, due to the changes in many properties of carbon materials such as optics and electricity [25,26], researchers believe that P can be used as a next-generation doping study of hetero elements. The current methods for preparing P-doped carbon mainly include: (1) post-treatment of carbon materials with P-containing substances such as phosphoric acid; (2) blending P-containing precursors with carbon-containing precursors and calcining them; and (3) vapor deposition using P-containing and carbon-containing precursors material [27,28]. In general, the use of the above methods has a certain amount of resource loss, which is not conducive to sustainable development. In fact, P-doped porous carbon materials can be directly obtained using biomass as both a P source and a carbon source due to phosphorus being a ubiquitous element in nature [29,30]. Now, P doping of carbon materials can change the conductivity and band gap of carbon materials for more efficient photocatalytic oxidation reactions [31,32]. However, there are few reports on the photocatalytic hydrogen production system, and further development is needed.

Pollen is a biomass precursor with unique surface folds that can improve the utilization of light, and pollen carbon is rich in N and P elements, which can be retained in the pollen carbon matrix during the carbonization process [33,34]. The existence of N, P elements can effectively change the energy band structure of pollen carbon, and at the same time can improve its electrical conductivity and photocatalytic activity [35,36]. In recent years, there have been some studies using pollen as a template to prepare adsorbents, catalysts, and energy storage materials [37].

In this work, nitrogen and phosphorus content-rich lotus pollen was selected as the bio-inspired template. The N and P elements self-doped porous carbon materials were obtained by a two-step calcination treatment, which can significantly improve the H_2_ production performance of it under the visible light irradiation. In addition, we found that the pollen carbon samples annealed at 600 °C exhibited unique photo-electrochemical (PEC) properties. With the increase in the applied bias voltage, the incident monochromatic photon current conversion efficiency (IPCE) of the sample increased sharply; meanwhile, its bandgap was narrowed significantly. Similar behavior was not observed in the carbon black and in those samples annealed at other temperatures. It is envisioned that this work will provide a new perspective for the contribution of the biochar materials to photocatalytic H_2_ production. It may also provide new ideas for the design and development of new biomass carbon materials and their applications in the field of photocatalysis/photoelectrochemical H_2_ generation.

## 2. Materials and Methods

### 2.1. Preparation of Pollen Carbon Material

The preparation process of pollen carbon materials is shown in Figure 1. Firstly, the purchased lotus pollen (20 g) was pretreated by immersion in ethanol solution (200 mL) to remove amino acids, proteins, and nucleic acids on the surface of the pollen, then dried in an oven at 60 °C for 12 h, marked as LP-Et. Secondly, the morphology of pollen was fixed by calcinating the LP-Et in a muffle furnace at a heating rate of 5 °C/min at 300 °C for 6 h. Thirdly, the morphology-fixing material was calcined in an argon atmosphere at a heating rate of 10 °C/min for 3 h. Finally, LP-C with different calcination temperatures (500–800 °C) were obtained, which were named as LP-C-500, LP-C-600, LP-C-700, and LP-C-800, respectively.

### 2.2. Characterization

The morphology and composition of pollen carbon materials were observed with a field emission a field emission scanning electron microscope (FESEM) (Apreo S Lo Vac, CZ, Thermo Fisher, Waltham, MA, USA) and a high-resolution transmission electron microscope (HRTEM) (Talos F200X, Thermo Fisher, Waltham, MA, USA, 200 kV voltage). The specific surface areas of materials were characterized by an automatic gas adsorption instrument (AutoSorb-iQ, Quantachrome, Boynton Beach, FL, USA) using N_2_ as adsorbent and the Brunauer–Emmett–Teller (BET) method. The X-ray diffractometer (XRD) system (Miniflex 600, Akishima, Rigaku, Tokyo, Japan) was used to characterize the crystal phase and structure of the synthesized materials. Additionally, the test was performed in the 2θ range of 10–90° with a scan rate of 5°/min with Cu Kα (λ = 0.15406 nm) Scan. Raman spectroscopy (DXR 2Xi, Thermo Fisher, Waltham, MA, USA) was used to study the structure of pollen carbon materials. Fourier transform infrared (FT-IR) spectra were measured by an infrared spectrophotometer (Nicolet iS 50, Thermo Fisher, Waltham, MA, USA). The surface elements of the prepared materials were analyzed on X-ray photoelectron spectroscopy (XPS, Scientific K-Alpha, Thermo Fisher, Waltham, MA, USA) with Al Kα radiation, and all binding energies were calibrated by the C1s peak at 284.8 eV.

### 2.3. Photocatalytic (PC) Performances

The photocatalytic hydrogen production performance of the samples was characterized by the MCP-WS1000 photochemical workstation (Beijing Perfectlight Technology Co., Ltd., Beijing, China). Typically, 10 mg of each photocatalyst was added into 30 mL of an aqueous solution which containing 50% volume of ethanol as a sacrificial agent (pH = 7), which were deaerated by vacuuming before H_2_ evolution measurement. The temperature of the solution was then controlled at 5 °C using a water-cooling system for the photocatalytic reaction. The full spectrum source is a simulated sunlight consisting of 9 LED lamps (365 nm, 385 nm, 420 nm, 450 nm, 485 nm, 535 nm, 595 nm, 630 nm, and one white light LED which is 420–750 nm) and the power of the total light irradiation was around 100 mW·cm^−2^. Additionally, the generated H_2_ was periodically analyzed using the PLD-CGA1000 compound gas analyzer (Beijing Perfectlight Technology Co., Ltd., Beijing, China).

### 2.4. Electrochemical (EC) and Photoelectrochemical (PEC) Performance

All of the electrochemical and photoelectrochemical measurements were carried out in a standard quartz made three-electrode cell. Among them, pollen carbon electrode, Pt foil and Ag/AgCl electrode (saturated KCl) were used as working electrode, counter electrode and reference electrode, respectively. The 0.1 M Na_2_SO_4_ aqueous solution (pH = 6.5) was used as the supporting electrolyte, which were deaerated by bubbling high-purity Ar for 15 min before EC/PEC measurements.

The typical preparation process of a working electrode is as follows: 20 mg of each photocatalyst and 50 μL of Nafion solution were dissolved in 700 μL of ethanol, which was ultrasonically dispersed to form a homogeneous suspension. The suspension was then dropped onto a 1 cm^2^ FTO glass and dried overnight. The final weight of photocatalyst loaded on each FTO glass was around 1 mg.

Linear Sweep Voltammogram (LSV) and Chronoamperometry (I-t) measurements were carried out in a photoelectrochemical test system (PEC2000, Beijing Perfectlight Technology Co., Ltd., Beijing, China) which is connected with a conventional electrochemical workstation (CHI 760E, Shanghai Chenhua, Shanghai, China). A 300 W Xenon Lamp equipped with filter (AM 1.5G) and power density 100 mW cm^−2^ (PLS-SXE300D, Beijing Perfectlight Technology. Co., Ltd., Beijing, China) was used as an illumination source. Photocurrent ON/OFF cycles were measured using the PEC2000 test system coupled with a mechanical chopper.

Typically, LSV measurements were performed in the potential range of 0.0 to 1.2 V (vs. Ag/AgCl) with a scan rate of 5 mV s^−1^. Chronoamperometry measurements were conducted at 1.0 V (vs. Ag/AgCl) under intermittent or constant illumination. The measured potential vs. Ag/AgCl was converted to the reversible hydrogen electrode (RHE) according to the Nernst Equation (1):*E* (vs. RHE) = *E* (vs. Ag/AgCl) + 0.0591 × pH+ *E*^0^(Ag/AgCl).(1)
where *E*^0^ (vs. RHE) = 0.1976 V at 25 °C.

Moreover, the Mott–Schottky, the electrochemical impedance spectroscopy (EIS) and the monochromatic incident photon-to-electron conversion efficiency (IPCE) measurements were carried out in the IPCE1000 photoelectrochemical test system (Beijing Perfectlight Technology Co., Ltd., Beijing, China.). The electrochemical system was equipped with an electrochemical station (CS 350H) to detect the photocurrent density and carrier transport speed of the samples.

## 3. Results and Discussion

### 3.1. Characterizations of Catalysts

SEM analysis showed the microstructure and morphology of lotus pollen after carbonization. It can be seen from Figure 2a,b that the pretreated pollen is spherical particles with a diameter of about 50 μm, and the surface presents a porous interconnected network pore structure. As shown in Figure 2c–f, the carbonized pollen retained the interconnection network structure and the size of it become smaller. This is because high temperature calcination removes some impurities while shrinking the particles, but the morphology and structure of the pollen remain unchanged during the carbonization process. In addition, the element mapping images in Figure 2g–j shows the presence of C, N, O, and P elements in pollen carbon. TEM further observed the morphology of LP-C-600 and its element mapping images (Figure 3), which further illustrated that the uniform distribution of C, N, O, and P elements in LP-C-600.

As shown in Figure 4a, there is no obvious crystalline peak in the XRD pattern of the pollen carbon materials, and the two broad peaks near 26.3° and 44.3° correspond to the (002) and (101) crystal planes of carbon, respectively. This shows that the pollen carbon photocatalyst was successfully prepared, and the materials were amorphous carbon.

We also used the Raman spectroscopy to study the effect of N, P self-doping on the structure of pollen carbon materials. As shown in Figure 4b, there are two peaks at the Raman shifts of 1336.2 cm^−1^ and 1588.8 cm^−1^, representing the D-peak and G-peak, respectively. The D-peak is attributed to the defect sites of graphite or disordered sp^2^-hybridized carbon atoms, and the G-peak represent the vibrations of sp^2^-bonded carbon atoms in a two-dimensional hexagonal lattice [38]. This corresponds to the XRD test result, which further illustrates that the prepared catalyst is an amorphous biomass carbon material.

Furthermore, the porous structure and specific surface area of the samples were analyzed through nitrogen adsorption–desorption experiments. As shown in Figure 4c, it was found that the samples all showed unclosed isotherms. This may be because the pore diameter shrinks after the biochar materials adsorb the gas, and the gas cannot be desorbed. As shown in Figure 4d, the pore size distribution analysis shows that there are micropores with a diameter of 0.5 nm in LP-C-800, while LP-C-500, LP-C-600, and LP-C-700 are mainly mesoporous structures. We also analyzed in detail the pore structure parameters of pollen carbon materials (Table 1). It can be seen that LP-C-800 has the highest specific surface area (350.454 m^2^/g), and LP-C-600 (81.685 m^2^/g) is second only to it. This is mainly because the porosity and specific surface area of bio-carbon materials are greatly affected by the pyrolysis temperature. The specific surface area of biochar increases dramatically when the pyrolysis temperature increases to 750 °C and above. Higher pyrolysis temperatures also cause the walls between adjacent pores to crack, resulting in higher porosity and increased surface area [39].

The functional groups on the surface of pollen carbon materials are characterized by FTIR as shown in Figure 4e. For pollen carbons, the peak at 1030 cm^−1^ was assigned to the stretching mode of hydrogen-bonded P=O, which indicates that the self-doping form of P in pollen carbons is mainly to form a chemical bond with oxygen. The appearance of absorption peaks near 1380 cm^−1^ and 3440 cm^−1^ are related to the -OH stretching vibration caused by the physical adsorption of water, and the absorption peak at 1650 cm^−1^ corresponds to the C=C in the samples [40]. The XPS measurement further confirmed the existence of the C, N, O, and P elements in pollen carbon (Figure 4f). It can be seen that the contents of P and N in pollen carbon materials prepared at different calcination temperatures are different. Among which, the relative content of P in LP-C-600 is the highest (1.95 at%), the relative content of N in LP-C-500 is the highest (7.07 at%), and the carbon black (CB) contains almost no N and P element.

In order to better clarify the existence form of each element in the biochar samples, we have conducted an in-depth exploration of the element state in the samples. From the C1s high-resolution spectrum shown in Figure 5a, it can be seen that each sample has fitting peaks around 284.8 eV and 286 eV. These two fitted peaks are associated with the presence of C-C and C-O-C, respectively, indicating that there are some carbon-related groups in both CB and pollen carbon materials. In addition, C-P (around 284.0 eV) and C=O (around 286.5 eV) bonds exist in the pollen carbon material, but not in CB [41]. As shown in Figure 5b, the characteristic peak of N does not appear in the CB sample, while the three fitting peaks of N 1s in LP-C-500, LP-C-600, LP-C-700, and LP-C-800, respectively located near 398 eV, 399.9 eV, and 402.9 eV, representing pyridine N, graphite N, and oxide N [42]. The O 1s high-resolution spectrum of CB in Figure 5c is divided into three peaks near 532.4 eV, 533.8 eV, and 534.1 eV, which are derived from the presence of lattice oxygen C-O, C=O, and -OH. However, the pollen carbon samples LP-C-500, LP-C-600, LP-C-700, and LP-C-800 have two fitting peaks at 530.5 eV and 532.1 eV, corresponding to lattice oxygen and C-O. Furthermore, it can be seen from Figure 5d that there is no P element in CB, while the XPS spectrum of P 2p of LP-C-500, LP-C-600, LP-C-700, and LP-C-800 can be deconvoluted into two peaks near 132.6 eV and 134.2 eV, corresponding to the P-C, and P-O bonds, respectively. The two fitted peaks correspond to P-C and P-O bonds, which correspond to the previous FT-IR analysis results, further explaining the role of P in pollen carbon [43].

Through the above specific characterization and analysis, it can be seen that the pollen carbon materials were doped with N, P elements. The N element mainly existed in the form of pyridine N, graphite N and oxide N, and the P element existed in the form of P-O bond and P-C bond. However, there are almost no N and P elements in carbon black.

### 3.2. Photocatalytic Activity

In order to gradually explore the specific effect of phosphorus on the photocatalytic performance of the material, we first conducted a photocatalytic water decomposition test for hydrogen production. The obtained pollen carbon material and commercial carbon black (CB) were tested for hydrogen production performance under full spectrum test conditions for 10 h. As shown in Figure 6a, the hydrogen production of pollen carbon materials is significantly higher than that of CB, and LP-C-600 has the highest hydrogen production activity (3.5 μmol·g^−1^·h^−1^). In addition, we also compared the hydrogen production performance of CB and LP-C-600 under visible light and full spectrum test conditions. As shown in Figure 6b, we found that the hydrogen production of CB and LP-C-600 was almost the same under the two test conditions, which indicated that the pollen carbon material also had obvious hydrogen production activity under visible light conditions. Therefore, we further tested the hydrogen production stability of LP-C-600 under full spectrum condition. As shown in Figure 6c, we found that it can always maintain a relatively constant hydrogen production rate during repeated tests, indicating that its catalytic activity is relatively stable.

Next, we conducted a series of electrochemical and photoelectrochemical tests to better understand the synergistic effect of P self-doping on the photocatalytic activity of pollen carbon materials. It can be seen from Figure 7a that, under AM 1.5G illumination, the photocurrent density of pollen carbon materials was significantly better than that of CB. Among several pollen carbon materials, LP-C-600 has the highest photocurrent density. In addition, it can be seen from Figure 7b that the photoelectric conversion efficiency of LP-C-600 is also improved several times compared with other samples. When the bias voltage reached 0.8 V (vs. Ag/AgCl), the photocurrent density achieved 5.8 μA/cm^2^, indicating that LP-C-600 has a higher charge separation efficiency. Figure 7c shows that, under the test conditions with an illumination interval of 300 s, the pollen carbon material still maintains a higher photocurrent response than CB during repeated light switching. The photocurrent density of LP-C-600 is still the highest, indicating that it has more stable photocatalytic water splitting performance compared with other samples [44].

As shown in Figure 8a, we measured the monochromatic photocurrent response of CB and pollen carbon materials under electrochemical noise mode (without any bias). As can be seen from the analysis results, CB has almost no photocurrent in the visible and ultraviolet regions, while the photocurrents of LP-C-500, LP-C-700, and LP-C-800 gradually disappear at the wavelength of 400 nm. However, LP-C-600 has obvious photocurrent in the wavelength range of 300–600 nm, indicating that it has photocurrent response in the ultraviolet and visible light regions. The photocurrents density curves shown in Figure 8b show that the photocurrent response of the samples is improved after the bias voltage is applied, and the photocurrent also increases with the increase in the bias voltage.

The function of wavelength (*k*) can be used to calculate the incident monochromatic photon current conversion efficiency (IPCE), as shown in Equation (2).
IPCE (%) = 1240 × [*I_ph_*/(*k* × *P_in_*)] × 100% (2)
where *P_in_* is the optical power density of incident light with wavelength (W·cm^−2^), and *I_ph_* is the output photocurrent density (A·cm^−2^). As shown in Figure 8c, under unbiased conditions, the IPCE (%) of LP-C-600 at 350 nm is 0.05%. Additionally, from Figure 8d, it can be seen that under the 0.8 V bias condition, the IPCE (%) of LP-C-600 at 350 nm is 0.11%, which is two times higher than that of the unbiased conditions. Furthermore, we found that the photo response of LP-C-600 sharply increased with the increase in the applied bias voltage. This was not observed for pollen carbon and carbon black materials at other annealing temperatures, revealing that the optimal ratio of doping element to carbon element in LP-C-600 was achieved.

From the Mott–Schottky curves in Figure 9a, it can be seen that the flat band potentials of CB, LP-C-500, LP-C-600, LP-C-700, and LP-C-800 are −0.01 V, −0.12 V, −0.134 V, −0.124 V, and −0.127 V, respectively. The flat band potential is usually the semiconductor conduction band position. The experimental results show the conduction band potential of LP-C-600 is more negative than CB, indicating that LP-C-600 has a stronger electron reduction ability and is more conducive to the photocatalytic hydrogen production reaction.

In addition, the electrochemical impedance spectroscopy (EIS) of samples is shown in Figure 9b. It can be seen from the Figure that, compared with the pollen carbon material, the EIS of CB has a smaller semicircular radius, which indicates that the electronic resistance of CB is smaller and the conductivity is better. However, there is no relevant literature that clearly shows an obvious linear correlation between the conductivity of the sample and the photoelectrochemical performance. Therefore, the conductivity results of the samples can only be used as a simple reference. As shown in Figure 9c, the band gap of the samples can be evaluated by (IPCE % × *hν*)^1/2^ and photon energy (*hν*) [45], and the bandgap extraction values for LP-C-500, LP-C-600, LP-C-700 and LP-C-800 were found to be 2.94 eV, 2.97 eV, 3.02 eV, and 3.07 eV, respectively.

Table 2 summarizes the hydrogen production performance and specific test conditions of previously reported bio-templated photocatalytic materials. It is worth noting that there are many factors that affect the hydrogen evolution rate, such as catalyst dosage, optical power density, cocatalyst, and reaction solvent [46]. Therefore, the photoelectrochemical performance of each material cannot be accurately compared by only the hydrogen production performance, and indicators such as incident monochromatic photon current conversion efficiency (IPCE) of samples should also be comprehensively considered.

Qian et al. fabricated porous carbon tubes embedded with CeO_2_ quantum dots (QDs) using biomass bean sprouts as a template. This three-dimensional porous Nano-CeO_2_/carbon material exhibits excellent hydrogen production performance (120 μmol·g^−1^·h^−1^) [47]. Cha et al. prepared a composite catalyst Pt/CdS/CSC with CdS quantum dots and Pt nanoparticles using coconut shell carbon (CSC) nanosheets, and the hydrogen production rate was as high as 1679.5 μmol·g^−1^·h^−1^, but hydrogen evolution was not observed in Pt/CSC without CdS quantum dots [48]. Luhong Zhang et al. mixed kapok fiber (KF) with melamine to produce carbon-modified graphitic carbon nitride (1.5CCN650) by one-step pyrolysis, and its hydrogen production rate was 1889 μmol·g^−1^·h^−1^ [49]. Jingtao Zhang et al. prepared a novel carbon fiber (CF) and g-C_3_N_4_ composite photocatalyst with a hydrogen production rate of 1080 μmol·g^−1^·h^−1^ [50]. Luhong Zhang et al. prepared carbon black-modified graphitic carbonitrides (Carbon black/g-C_3_N_4_) with a simple hybrid thermal strategy, and its hydrogen production rate was 234 μmol·g^−1^·h^−1^ [51].

It can be seen that LP-C-600 still exhibits obvious photocatalytic activity without co-catalysts such as noble metals and with low optical power density. Moreover, from the previously discussed visible light response range and external quantum efficiency (IPCE) and band gap values, pollen carbon materials have obvious advantages in photoelectric catalytic scenarios. In conclusion, compared with other noble metal-modified biomass photocatalysts, pollen carbon materials show great potential both in terms of environment and cost.

Finally, based on the above results, we propose the band structure of as prepared pollen carbon materials (Figure 10).

## 4. Conclusions

In summary, the N, P self-doped lotus pollen like-carbon with a hierarchical pore structure was successfully prepared through a two-step calcination method. Meanwhile, its photocatalytic hydrogen production performance has been studied. The results show that the prepared pollen carbon materials exhibit better performance than carbon black in photocatalytic hydrogen production and photoelectrochemical tests. Among them, LP-C-600 has the best hydrogen production performance (the hydrogen generation rate reaches 3.5 μmol·g^−1^·h^−1^) and the highest photocurrent (5.8 μA·cm^−2^) under 0.8 V bias. In addition, LP-C-600 has a very wide photo response range (300–600 nm), and its IPCE value reaches the highest value of 0.11% at 350 nm under 0.8V bias. Furthermore, the visible light assisted PEC performance of the LP-C-600 sharply increases with the increase in the applied bias voltage. This could be attributed to the surface wrinkles and N, P self-doped of the pollen carbon materials, which are beneficial to improve the light utilization efficiency, and the generation and transfer of photo-generated carriers.

This work demonstrates a strategy to obtain more efficient photocatalysts based on biomass, which not only broadens the application of biochar materials in the field of photocatalysis/photoelectrochemistry, but also provides new ideas for the design and development of novel biomass carbon materials.

## Figures and Tables

**Figure 1 nanomaterials-12-01744-f001:**
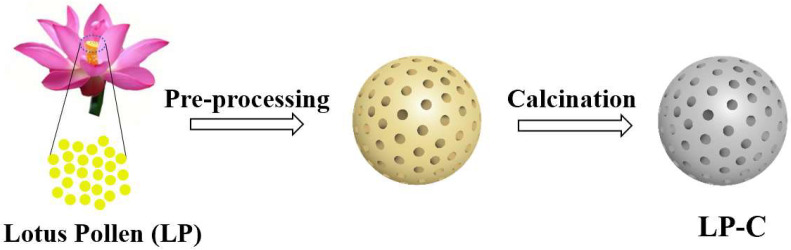
Schematic diagram of the synthetic route of pollen carbon materials.

**Figure 2 nanomaterials-12-01744-f002:**
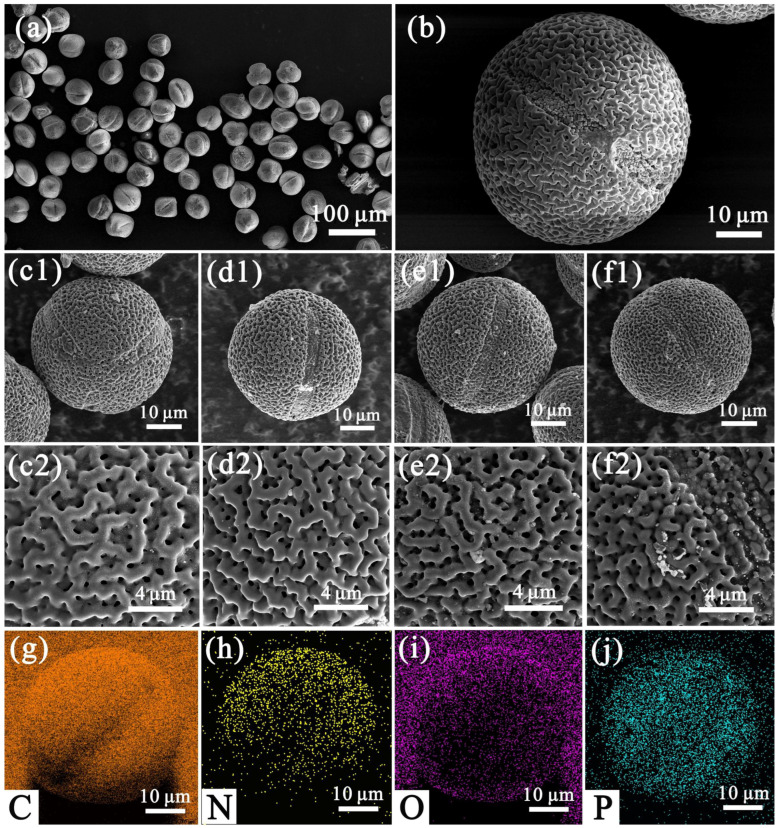
SEM images of (**a**,**b**) LP-Et, (**c1**,**c2**) LP-C-500, (**d1**,**d2**) LP-C-600, (**e1**,**e2**) LP-C-700, (**f1**,**f2**) LP-C-800; and (**g**–**j**) element mapping images of C, N, O, and P in LP-C-600.

**Figure 3 nanomaterials-12-01744-f003:**
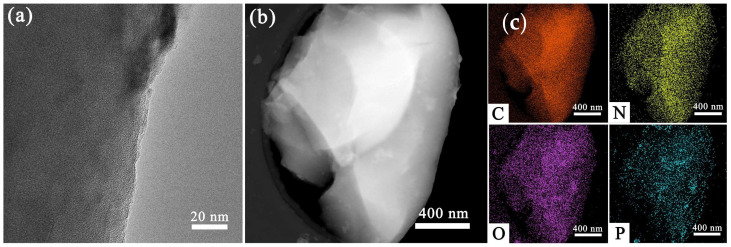
(**a**) High-magnification TEM image, (**b**) low-magnification TEM image and (**c**) element mapping images of C, N, O, and P.

**Figure 4 nanomaterials-12-01744-f004:**
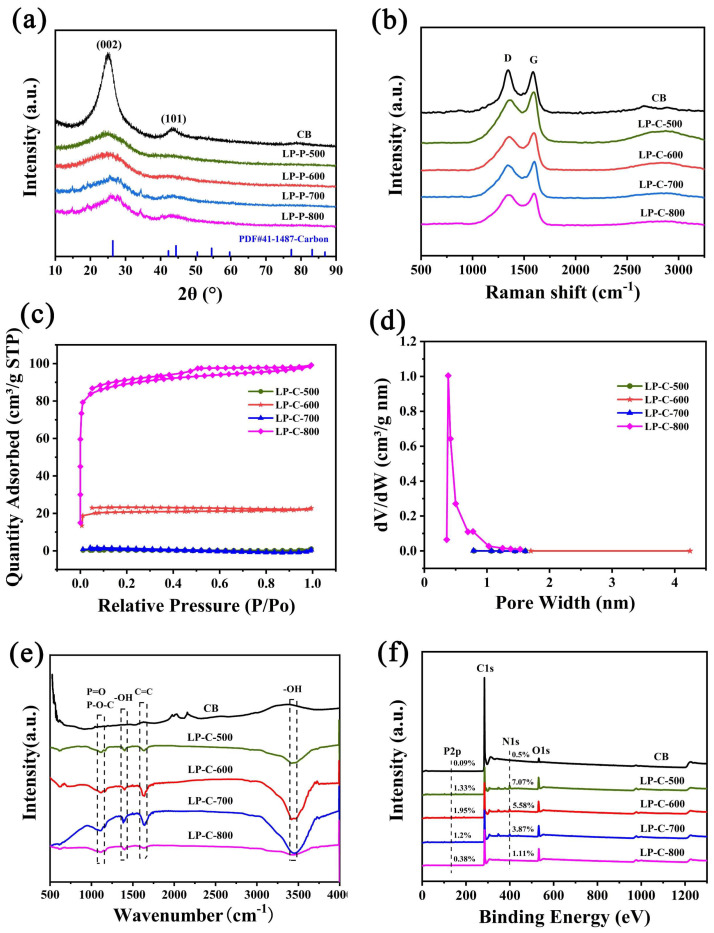
(**a**) X-ray diffraction patterns, (**b**) Raman spectroscopy, (**c**) N_2_ adsorption–desorption isotherms; (**d**) pore size distribution, (**e**) FT-IR spectroscopy, and (**f**) XPS spectra of samples.

**Figure 5 nanomaterials-12-01744-f005:**
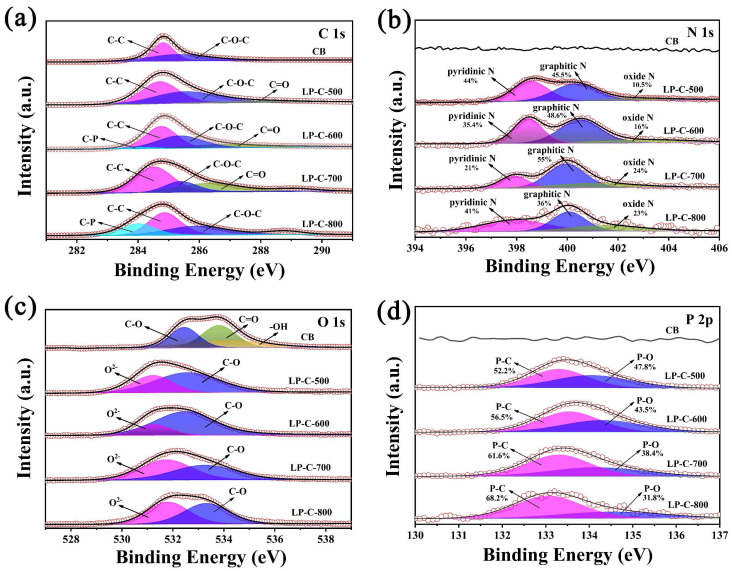
High-resolution XPS spectra of (**a**) C 1s, (**b**) N 1s, (**c**) O 1s, and (**d**) P 2p of samples.

**Figure 6 nanomaterials-12-01744-f006:**
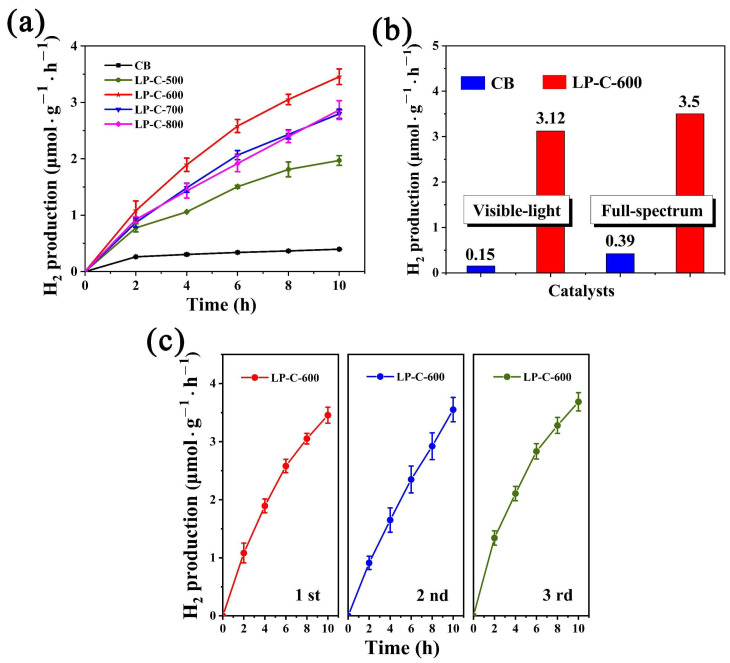
(**a**) H_2_ evolution performance of samples, (**b**) comparison of H_2_ evolution performance of samples under visible light and full spectrum, and (**c**) H_2_ evolution stability of LP-C-600.

**Figure 7 nanomaterials-12-01744-f007:**
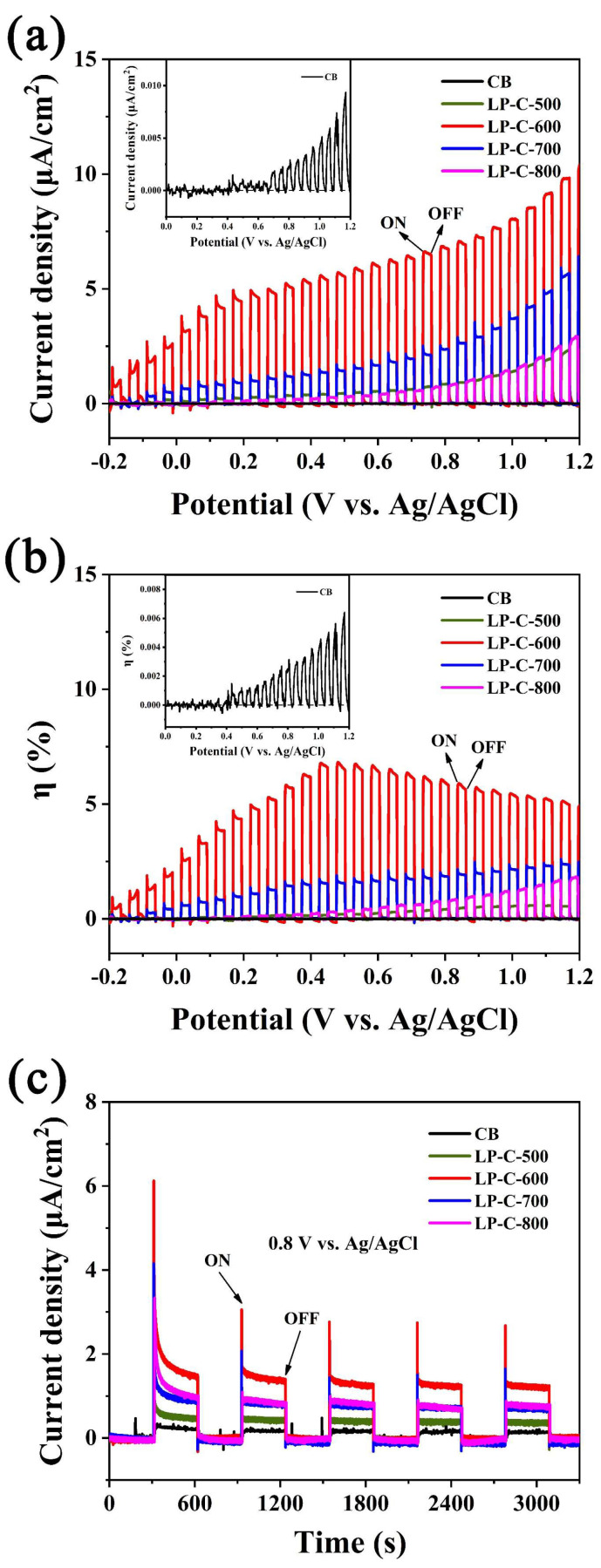
(**a**) Linear sweep voltammogram (LSV) curves, (**b**) photoelectric conversion efficiency, and (**c**) photocurrent density time plot at applied voltage of 0.8 V of samples under test conditions with 0.1 M Na_2_SO_4_.

**Figure 8 nanomaterials-12-01744-f008:**
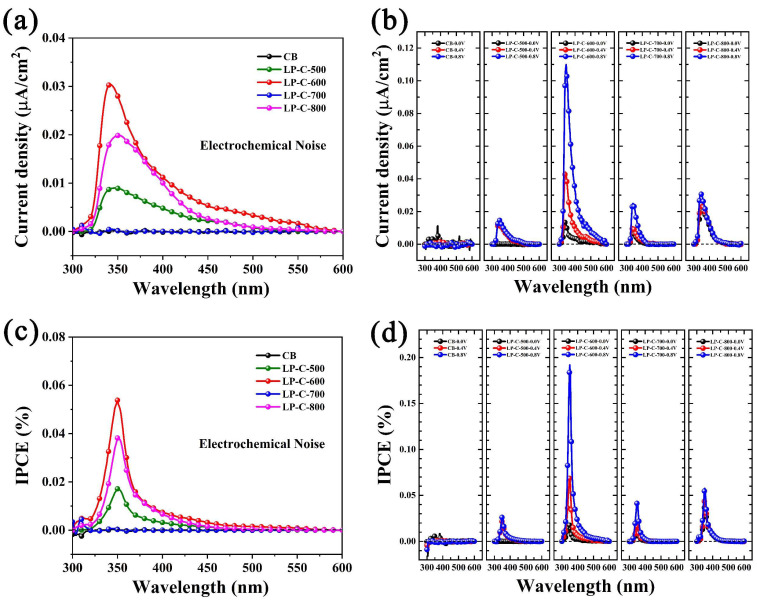
Photocurrents density of samples versus the wavelength of incident light at (**a**) electrochemical noise mode and (**c**) with 0 V, 0.4 V, and 0.8 V applied voltages (vs. Ag/AgCl); (**b**,**d**) the derived IPCE (%) spectra.

**Figure 9 nanomaterials-12-01744-f009:**
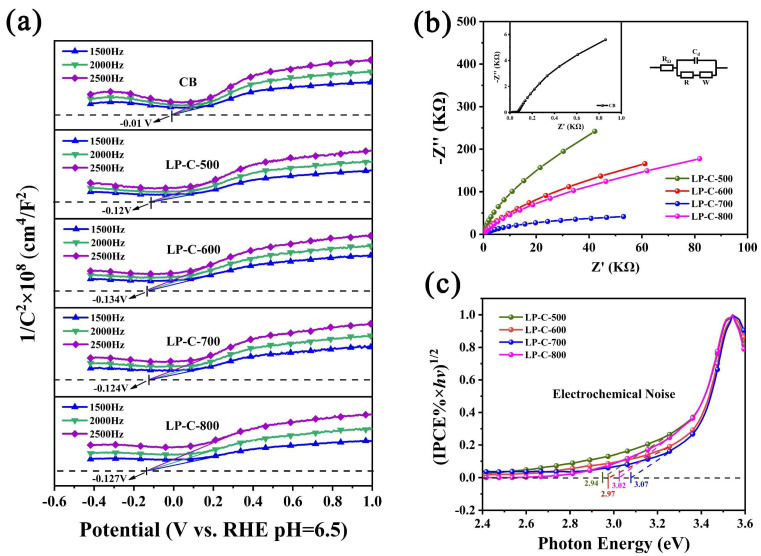
(**a**) Mott−Schottky plots, (**b**) electrochemical impedance spectroscopy (EIS), and (**c**) the band gap determination extracted from IPCE spectra by a function of (IPCE % × *hν*)^1/2^ vs. photon energy of samples.

**Figure 10 nanomaterials-12-01744-f010:**
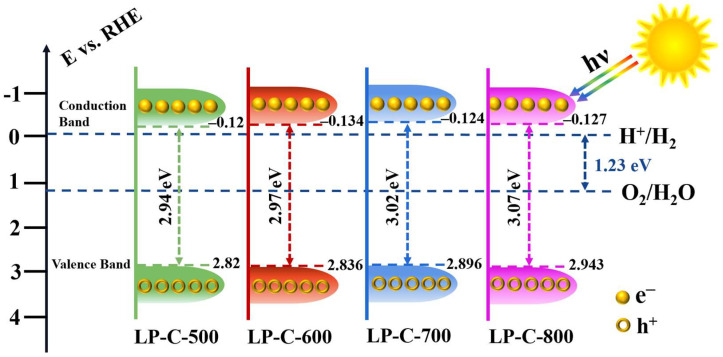
Proposed band structure of pollen carbon and redox potentials of water splitting.

**Table 1 nanomaterials-12-01744-t001:** Pore structure parameters from N_2_ adsorption of samples.

Samples	S_BET_ (m^2^/g)	S_mic_ (m^2^/g)	V_t_ (cm^3^/g)	V_mic_ (cm^3^/g)	V_mic_/V_t_
LP-C-500	1.039	0.612	0.000238	0.000236	0.991
LP-C-600	81.685	76.066	0.033854	0.029557	0.873
LP-C-700	2.353	0.885	0.004092	0.003322	0.871
LP-C-800	350.454	298.074	0.150731	0.114759	0.761

**Table 2 nanomaterials-12-01744-t002:** Reported hydrogen evolution rates and specific test conditions for bio-templated photocatalytic materials.

Samples	Co-Catalyst	Photocatalyst Dosage (mg)	Solvent	Reaction Parameters	H_2_ Production Activity (μmol·g^−1^·h^−1^)	Photoresponse Cut-off Wavelength/Band Gap	Ref.
Nano-CeO_2_/carbon	Nano-CeO_2_	100	MeOH/H_2_O	300 W Xe lamp	120	Null/3.08 eV	[47]
Pt/CSC	Pt	100	20 vol% Lactic acid	300 W Xe lamp/visible light	0	Null	[48]
1.5CCN650	Pt	10	TEOA/H_2_PtCl_6_/H_2_O	Xe lamp/λ > 420 nm	1889	Null	[49]
CF-C_3_N_4_	Pt	50	TEOA/H_2_O	350 W Xe lamp/ 180 mW/cm^2^	1080	Null	[50]
Carbon black/g-C_3_N_4_	Pt	10	TEOA/H_2_O	Null	234	Null	[51]
Carbon Black	Null	10	EtOH/H_2_O	LED light/100 mW/cm^2^	0.4	300 nm/Null	This work
LP-C-600	Null	10	EtOH/H_2_O	LED light/100 mW/cm^2^	3.5	600 nm/2.97 eV	

## Data Availability

Data is contained within the article.

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
