# Peer review of "N, P Self-Doped Porous Carbon Material Derived from Lotus Pollen for Highly Efficient Ethanol–Water Mixtures Photocatalytic Hydrogen Production"

_nanomaterials, 2022, doi:10.3390/nano12101744_

Round 1

Reviewer 1 Report

This manuscript entitled “N, P Self-doped Porous Carbon Material Derived from Lotus  Pollen for Highly Efficient Ethanol-Water Mixtures Photocatalytic Hydrogen Production” by Jing-Wen Zhou et al., have prepared variety of the pollen carbon photocatalyst for hydrogen production. Such study must be helpful towards greener energy production, which has been a worldwide issue. It can be publishable on Nanomaterials after a minor revision after addressing following important points:

  1. Line 91 Authors claimed “inter-layer voids and adsorption position to improve conductivity” How interlayer voids can improve conductivity? Kindly explain.
  2. Line 204-205 “The D-peak is ascribed to the sp3 bond carbon atom, and the G-peak represents the sp2 hybridization carbon atom in amorphous carbon [41].” This cannot be the right explanation for Raman D and G bands, as they correspond to any graphitic structure, crystalline or amorphous. Kindly improve this.
  3. Kindly explain why for LP-C-800 the porosity is tremendously high compared to other samples? What is happening at 800oC in particular, mechanism of carbonization should be discussed.
  4. Line 342-344 “ This could be attributed to the surface wrinkles, abundant pore structure and N, P self-doped of the pollen carbon materials, which are beneficial to improve the light utilization efficiency” The LP-C-800 which has highest porosity and surface area is not the best photocatalyst here, so abundant pores can not be related to the performance.
  5. Language improvement can improve the readability of the article. On several instances sentence splitting is necessary. For example : Line 17-20 “In this work, lotus pollen (LP) was used as the carbon source, a variety of the pollen carbon photocatalyst were prepared by a two-step roasting method. A series of characterizations were per-18 formed on the as prepared samples, which with an average particle size of about 40 μm, exhibited a typical structure of amorphous carbon and a network-like interconnected porous structure on the surface.”, Line 34-35 “In this case, there is an urgent need for researchers to develop renewable green energy sources [2-3], and hydrogen energy has… can be split into two statement, Line 72-74, Line 93-98 “In this work, we selected 93 lotus pollen with higher phosphorus content as N and P source, lotus pollen-like N, P-94 self doped porous hollow carbon material was obtained through two-step calcination, the effect of roasting temperature on the properties of the material were explored, and the structure-activity relationship between the obtained material and hydrogen production by photocatalytic hydrolysis was investigated.” Van be split into two sentences… etc
  6. Authors should explain why this work is necessary, if they reports very low H2 production compared to any other report in the literature?

Author Response

Response to Reviewers

Nanomaterials

Manuscript ID: nanomaterials-1707186

Authors: Jing-Wen Zhou, Xia Jiang, Yan-Xin Chen*, Shi-Wei Lin, Can-Zhong Lu*

Dear editor:

We appreciate very much for your constructive comments and thoughtful suggestions on our manuscript. Those comments are valuable and very helpful. According to your suggestions, we have carefully reviewed the comments and made revision on the original manuscript. All revised sections were marked in red in the revised manuscript which we would like to submit for your kind consideration. The responses to the reviewer's comments are presented as following:

Reviewer #1 Comments: This manuscript entitled “N, P Self-doped Porous Carbon Material Derived from Lotus Pollen for Highly Efficient Ethanol-Water Mixtures Photocatalytic Hydrogen Production” by Jing-Wen Zhou et al., have prepared variety of the pollen carbon photocatalyst for hydrogen production. Such study must be helpful towards greener energy production, which has been a worldwide issue. It can be publishable on Nanomaterials after a minor revision after addressing following important points:

Comment 1: Line 91 Authors claimed “inter-layer voids and adsorption position to improve conductivity”. How interlayer voids can improve conductivity? Kindly explain.

Response 1: We are grateful for the comment. The original sentence in the paper involved in this comment is “Pollen is a biomass precursor owned hierarchical structure and rich in N and P, those can be retained in the pollen carbon matrix during the carbonization process, effectively changing the distribution of defects in the pollen carbon structure, interlayer voids and adsorption position to improve conductivity and photocatalytic activity.” This is to express that the N, P self-doping phenomenon in natural pollen materials can optimize the energy band structure of pollen carbon materials, and at the same time improve its electrical conductivity and photoelectric catalytic activity. So, it may not mean that “inter-layer voids and adsorption position to improve conductivity”. This may be due to some problems in the language expression of this sentence, which caused some misunderstandings. We have improved this sentence to “Pollen is a biomass precursor with a hierarchical structure and rich in N, P elements, which can be retained in the pollen carbon matrix during carbonization. The existence of N, P elements can effectively change the energy band structure of pollen carbon, and at the same time can improve its electrical conductivity and photocatalytic activity”. (Lines 68-72, page 2)

Comment 2: Line 204-205 “The D-peak is ascribed to the sp3 bond carbon atom, and the G-peak represents the sp2 hybridization carbon atom in amorphous carbon [41].” This cannot be the right explanation for Raman D and G bands, as they correspond to any graphitic structure, crystalline or amorphous. Kindly improve this.

Response 2: Thanks for pointing out. We have corrected the explanation in the paper to “Raman shifts of 1336.2 cm-1 and 1588.8 cm-1, representing the D-peak and G-peak, respectively. The D-peak is attributed to the defect sites of graphite or disordered sp2-hybridized carbon atoms, and the G-peak represent the vibrations of sp2-bonded carbon atoms in a two-dimensional hexagonal lattice”. (Lines 192-195, page 6)

Comment 3: Kindly explain why for LP-C-800 the porosity is tremendously high compared to other samples? What is happening at 800 ℃ in particular, mechanism of carbonization should be discussed.

Response3: Thanks for your kind comment. It is true that the porosity and specific surface area of biomass carbon materials are greatly affected by the pyrolysis temperature. Based on our experience and other literature reports, at low temperatures, the obtained biochar typically has a surface area of less than 10 m2/g. When the temperature is raised to 750 °C and above, the specific surface area of biochar increases sharply to around 400 m2/g. Higher pyrolysis temperatures also cause the walls between adjacent pores to crack, resulting in higher porosity and increased surface area [39]. This is the reason why the porosity and specific surface area of LP-C-800 prepared at 800 ℃ are higher than other samples. To be more clearly describe the experimental results and in accordance with the reviewer concerns, we have also added more detailed explanations and relevant reference [reference 39] in the paper. (Lines 209-213, page 7)

[39] W. J. Liu, H. Jiang, H. Q. Yu. Development of Biochar-Based Functional Materials: Toward a Sustainable Platform Carbon Material. Chemical Reviews. 2015, 115, 12251-12285.

Comment 4: Line 342-344 “This could be attributed to the surface wrinkles, abundant pore structure and N, P self-doped of the pollen carbon materials, which are beneficial to improve the light utilization efficiency”. The LP-C-800 which has highest porosity and surface area is not the best photocatalyst here, so abundant pores can’t be related to the performance.

Response 4: Thanks for pointing out. From the characterization and performance test we can see that the effect of pore structure on the performance of photocatalysts is indeed not the most important factor. What we really want to describe is that the unique surface wrinkles and N, P self-doping phenomenon of pollen carbon materials are the main factors affecting the performance. We are sorry for the unclear language description, which caused some misunderstandings. In order to avoid unnecessary controversy, have improved this sentence to “This could be attributed to the surface wrinkles and N, P self-doped of the pollen carbon materials, which are beneficial to improve the light utilization efficiency, and the generation and transfer of photo-generated carriers”. (Lines 385-387, page 14)

Comment 5: Language improvement can improve the readability of the article. On several instances sentence splitting is necessary. For example: Line 17-20 “In this work, lotus pollen (LP) was used as the carbon source, a variety of the pollen carbon photocatalyst were prepared by a two-step roasting method. A series of characterizations were per-18 formed on the as prepared samples, which with an average particle size of about 40 μm, exhibited a typical structure of amorphous carbon and a network-like interconnected porous structure on the surface.”, Line 34-35 “In this case, there is an urgent need for researchers to develop renewable green energy sources [2-3], and hydrogen energy has… can be split into two statement, Line 72-74, Line 93-98 “In this work, we selected 93 lotus pollen with higher phosphorus content as N and P source, lotus pollen-like N, P-94 self-doped porous hollow carbon material was obtained through two-step calcination, the effect of roasting temperature on the properties of the material were explored, and the structure-activity relationship between the obtained material and hydrogen production by photocatalytic hydrolysis was investigated.” Van be split into two sentences… etc.

Response 5: Thanks for the valuable suggestion. According to your suggestion, we have carefully checked the article language and made necessary sentence splitting to make it more readable. (Lines 18-21, page 1; Lines 34-36, page 1; Lines 75-87, page 2; Lines 227-231, page 7; Lines 329-334, page 12, etc.)

Comment 6: Authors should explain why this work is necessary, if their reports very low H2 production compared to any other report in the literature?

Response 6: Thanks for the kind comment. There are several reasons for the lower hydrogen production rate of pollen carbon materials compared with the values reported in other literatures. For example, most of the materials reported in the literature have supported noble metals as cocatalysts, which undoubtedly increases the cost of material preparation; On the other hand, due to different experimental conditions in various literatures, such as light intensity and different sacrificial agent addition, a completely fair comparison of hydrogen production rate is very difficult [46].

Although our prepared pollen carbon material has relatively low hydrogen production efficiency, its visible light response range is very wide. As shown in the Figure 8. (a), in the wavelength range of 300-600 nm, the pollen carbon samples all have a relatively obvious light response, while carbon black (CB) has almost no response. In addition, we tested the hydrogen production performance of the samples under visible light irradiation. As shown in Figure 6. (b), we can see that the pollen carbon samples exhibit similar hydrogen production activity under visible light and the full spectrum irradiation. Which further confirmed the Pollen carbon has excellent visible light response properties. Therefore, we think it is meaningful to study the photocatalytic properties of pollen carbon. This not only provides ideas for the preparation of new photocatalysts, but also expands the application of biomass carbon materials.

Figure 8. (a) Photocurrents density of samples versus the wavelength of incident light at electrochemical noise mode.

Figure 6. (b) Comparison of H2 evolution performance of samples under visible light and full spectrum.

[46] Wang, Z., Li, C., Domen, K. Recent developments in heterogeneous photocatalysts for solar-driven overall water splitting. Chem. Soc. Rev. 2019, 48.

Once again, we thank you and the reviewers for the time you put in reviewing our paper. We feel that your suggestions to be very valuable and have carefully revised them. Looking forward to our revised manuscript meeting your expectations.

Sincerely,

Can-Zhong Lu

Reviewer 2 Report

The following issues must be addressed:

  1. Table 1 must be upgraded to include more relevant papers;
  2. Last sentence from Introduction must be improved to outline what is new and innovative in this work;
  3. Doping is a procedure which includes a reliable method to control the qualitative and quantitative parameters – this is not the case. Is not clear if there is any method to control the quantity of doping agents. If not, then we speak about contamination. The authors must clarify these aspects.
  4. The authors must explain in details why there are such big differences between SBET of the samples.
  5. The influence of photocatalyst dosage in hydrogen production will be interesting to be evaluated as well.
  6. Conclusions should contain some specific values which are representative for this work.

Author Response

Response to Reviewers

Nanomaterials

Manuscript ID: nanomaterials-1707186

Authors: Jing-Wen Zhou, Xia Jiang, Yan-Xin Chen*, Shi-Wei Lin, Can-Zhong Lu*

Dear editor:

We appreciate very much for your constructive comments and thoughtful suggestions on our manuscript. Those comments are valuable and very helpful. According to your suggestions, we have carefully reviewed the comments and made revision on the original manuscript. All revised sections were marked in red in the revised manuscript which we would like to submit for your kind consideration. The responses to the reviewer's comments are presented as following:

Reviewer #2 Comments: The following issues must be addressed:

Comment 1: Table 1 must be upgraded to include more relevant papers.

Response 1: Thanks for the valuable suggestion. We further investigated and analyzed relevant literature (see Lines 338-345, page 13 and Lines 349-364, page 13), improved Table 1 and moved it to a more appropriate position in the paper (page 13), renaming it as Table 2. The specific improvements are as follows:

Table 2. Reported hydrogen evolution rates and specific test conditions for bio-templated photocatalytic materials.

[47] Junchao Qian, Zhigang Chen, Hui Sun, Feng Chen, Xing Xu, Zhengying Wu, Ping Li, Wangjie Ge. Enhanced Photocatalytic H2 Production on Three-Dimensional Porous CeO2/Carbon Nanostructure. ACS Sustainable Chem. Eng. 2018, 6, 9691-9698.

[48] Da-Wei Zha, Liang-Fang Li, Yun-Xiang Pan, Jian-Bo He. Coconut shell carbon nanosheets facilitating electron transfer for highly efficient visible-light-driven photocatalytic hydrogen production from water. Int. J. Hydrog. Energy. 2016, 41, 17370-17379.

[49] L. H. Zhang, Z. Y. Jin, S. L. Huang, X. Y. Huang, B. H. Xu, L. Hu, H. Z. Cui, S. C. Ruan, Y. J. Zeng. Bio-inspired carbon doped graphitic carbon nitride with booming photocatalytic hydrogen evolution. Appl. Catal. B: Environ. 2019, 246, 61-71.

[50] Zhang, J., Huang, F. Enhanced visible light photocatalytic H2 production activity of g-C3N4 via carbon fiber. Appl. Surf. Sci. 2015, 358, 287-295.

[51] Zhang, L., Jin, Z., Lu, H., Lin, T., Ruan, S., Zhao, X. S., Zeng, Y.-J. Improving the Visible-Light Photocatalytic Activity of Graphitic Carbon Nitride by Carbon Black Doping. ACS Omega. 2018, 3, 15009-15017.

Comment 2: Last sentence from Introduction must be improved to outline what is new and innovative in this work.

Response 2: Thanks for your valuable suggestion. We have improved the information that this comment said in the text, and the revised content is “In this work, nitrogen and phosphorus content-rich lotus pollen was selected as the bio-inspired template. The N and P elements self-doped porous carbon materials were obtained by a two-step calcination treatment, which can significantly improve the H2 production performance of it under the visible light irradiation. In addition, we found that the pollen carbon samples annealed at 600 °C exhibited unique photo-electrochemical (PEC) properties. As the increase of the applied bias voltage, the incident monochromatic photon current conversion efficiency (IPCE) of the sample increased sharply meanwhile the bandgap of it was narrowed significantly. Similar behavior was not observed on the carbon black and on those samples annealed at other temperatures. It is envisioned that this work will provides a new perspective for the contribution of the biochar materials to photocatalytic H2 production. It may also provide new ideas for the design and development of new biomass carbon materials and their applications in the field of photocatalysis / photoelectrochemical H2 generation.” (Lines 75-87, page 2)

Comment 3: Doping is a procedure which includes a reliable method to control the qualitative and quantitative parameters - this is not the case. Is not clear if there is any method to control the quantity of doping agents. If not, then we speak about contamination. The authors must clarify these aspects.

Response 3: We deeply appreciate your suggestion. As you mentioned, the presence of dopants and their specific levels do have an important effect on the performance of the catalyst. If the quantity of doping agents can be strictly controlled, the performance of the catalyst can undoubtedly be further optimized. But unfortunately, in the pollen carbon material prepared in this paper, the quantity of doped elements cannot be completely controlled. Although the specific content of the elements existing in the pollen carbon material in the form of self-doping cannot be precisely controlled, we can adjust the content of the quantity of doping agents by changing the firing temperature to make it reach a suitable range.

As shown in Figure. 4(f) in the paper, the contents of P and N in the pollen carbon materials prepared at different calcination temperatures are different, which is the result of the adjustment. However, it is not possible to strictly control the quantity of doping agents to a certain value for the time being. We will explore this aspect more deeply in the follow-up work.

Figure. 4. (f) XPS spectra of samples.

Comment 4: The authors must explain in details why there are such big differences between SBET of the samples.

Response 4: Thanks for your comment. In fact, the porosity and specific surface area of biomass carbon materials are greatly affected by the pyrolysis temperature. At low temperatures, the obtained biochar typically has a surface area of less than 10 m2/g. When the temperature is raised to 750 °C and above, the specific surface area of biochar increases sharply to around 400 m2/g. Higher pyrolysis temperatures also cause the walls between adjacent pores to crack, resulting in higher porosity and increased surface area [39]. This is the reason why there are such big differences between SBET of the samples. To be more clearly describe the experimental results and in accordance with the reviewer concerns, we have also added more detailed explanations in the paper. (Lines 209-213, page 7)

[39] W. J. Liu, H. Jiang, H. Q. Yu. Development of Biochar-Based Functional Materials: Toward a Sustainable Platform Carbon Material. Chemical Reviews. 2015, 115, 12251-12285.

Comment 5: The influence of photocatalyst dosage in hydrogen production will be interesting to be evaluated as well.

Response 5: Thanks for your kind suggestion. In fact, we have explored the effect of different dosages of the prepared pollen carbon photocatalysts on its hydrogen production performance during the pre-experiment. As shown in the figure below, when the catalyst dosage is 10 mg, the photocatalyst exhibit excellent hydrogen production performance. However, when the catalyst dosage increased to 20 mg, the hydrogen production rate decreased. This may be due to the excess catalyst makes the solution cloudy which blocking light transmission efficiency also increase the photogenerated carrier’s recombination rate. Therefore, in the subsequent repeated experiments, we selected 10 mg as the optimal amount of the catalyst.

Figure. Hydrogen production with different dosages of photocatalysts.

Comment 6: Conclusions should contain some specific values which are representative for this work.

Response 6: Thanks for the valuable suggestion. We have added specific values representing this work to the conclusion of the paper, improving the original conclusion to “Among them, LP-C-600 has the best hydrogen production performance (the hydrogen generation rate reaches 3.5 μmol·g-1·h-1) and the highest photocurrent (5.8 μA·cm-2 under 0.8 V bias. In addition, LP-C-600 has a very wide photo response range (300-600 nm), and its IPCE value reaches the highest value of 0.11% at 350 nm under 0.8V bias. Furthermore, the visible light assisted PEC performance of the LP-C-600 sharply increases with the increase of the applied bias voltage.” (Lines 379-385, page 14)

Once again, we thank you and the reviewers for the time you put in reviewing our paper. We feel that your suggestions to be very valuable and have carefully revised them. Looking forward to our revised manuscript meeting your expectations.

Sincerely,

Can-Zhong Lu

Reviewer 3 Report

In this work, the authors have studied the performance of m Lotus Pollen based Porous Carbon Material for the H2 production, in comparison with commercial carbon black (CB).The authors report that N, P self-doping was more effective than CB because of the unique properties. The work is interesting, however some comments should be addressed before its consideration:

  • The main issue is that Lotus Pollen has very low H2 production compared to reported materials as shown in Table 1. Even though, the mass of catalyst is lower, but a notable difference between reported values and the value of this work can be oberved.
  • Table 1 may contain more data. I suggest to report Table 1 at the end of the MS.
  • In real application, it is possible to collect enough quantity from based Porous for large scale application? Please discuss the avaialability and the porcessing costs. The authors should mention why they used this material.
  • The english should be checked carefully.

Author Response

--------------------------------------------------------------------------

Response to Reviewers

Nanomaterials

Manuscript ID: nanomaterials-1707186

Authors: Jing-Wen Zhou, Xia Jiang, Yan-Xin Chen*, Shi-Wei Lin, Can-Zhong Lu*

Dear editor:

We appreciate very much for your constructive comments and thoughtful suggestions on our manuscript. Those comments are valuable and very helpful. According to your suggestions, we have carefully reviewed the comments and made revision on the original manuscript. All revised sections were marked in red in the revised manuscript which we would like to submit for your kind consideration. The responses to the reviewer's comments are presented as following:

Reviewer #3 Comments: In this work, the authors have studied the performance of Lotus Pollen based Porous Carbon Material for the H2 production, in comparison with commercial carbon black (CB). The authors report that N, P self-doping was more effective than CB because of the unique properties. The work is interesting; however, some comments should be addressed before its consideration:

Comment 1: The main issue is that Lotus Pollen has very low H2 production compared to reported materials as shown in Table 1. Even though, the mass of catalyst is lower, but a notable difference between reported values and the value of this work can be observed.

Response1: We are extremely grateful to reviewer for pointing out this problem. There are several reasons for the lower hydrogen production rate of pollen carbon materials. For example, most of the materials reported in the literature listed in Table 2 have supported noble metals as cocatalysts, which undoubtedly increases the cost of material preparation; On the other hand, due to different experimental conditions in various literatures, such as light intensity and different sacrificial agent addition, a completely fair comparison of hydrogen production rate is very difficult [46]. In additional, although our prepared pollen carbon material has relatively low hydrogen production efficiency, its visible light response range is very wide. As shown in the Figure 8. (a), in the wavelength range of 300-600 nm, the pollen carbon samples all have a relatively obvious light response, while carbon black (CB) has almost no response. In addition, we tested the hydrogen production performance of the samples under visible light irradiation. As shown in Figure 6. (b), we can see that the pollen carbon samples exhibit similar hydrogen production activity under visible light and the full spectrum irradiation. Which further confirmed the Pollen carbon has excellent visible light response properties. Therefore, we think it is meaningful to study the photocatalytic properties of pollen carbon. This not only provides ideas for the preparation of new photocatalysts, but also expands the application of biomass carbon materials.

Figure 8. (a) Photocurrents density of samples versus the wavelength of incident light at electrochemical noise mode.

Figure 6. (b) Comparison of H2 evolution performance of samples under visible light and full spectrum.

[46] Wang, Z., Li, C., Domen, K. Recent developments in heterogeneous photocatalysts for solar-driven overall water splitting. Chem. Soc. Rev. 2019, 48.

Comment 2: Table 1 may contain more data. I suggest to report Table 1 at the end of the MS.

Response 2: Thanks for your valuable suggestion. We further investigated and analyzed relevant literature (see Lines 338-345, page 13 and Lines 349-364, page 13), improved Table 1 and moved it to your suggested location (page 13), renaming it as Table 2. The specific improvements are as follows:

Table 2. Reported hydrogen evolution rates and specific test conditions for bio-templated photocatalytic materials.

[47] Junchao Qian, Zhigang Chen, Hui Sun, Feng Chen, Xing Xu, Zhengying Wu, Ping Li, Wangjie Ge. Enhanced Photocatalytic H2 Production on Three-Dimensional Porous CeO2/Carbon Nanostructure. ACS Sustainable Chem. Eng. 2018, 6, 9691-9698.

[48] 49.   Da-Wei Zha, Liang-Fang Li, Yun-Xiang Pan, Jian-Bo He. Coconut shell carbon nanosheets facilitating electron transfer for highly efficient visible-light-driven photocatalytic hydrogen production from water. Int. J. Hydrog. Energy. 2016, 41, 17370-17379.

[49] L. H. Zhang, Z. Y. Jin, S. L. Huang, X. Y. Huang, B. H. Xu, L. Hu, H. Z. Cui, S. C. Ruan, Y. J. Zeng. Bio-inspired carbon doped graphitic carbon nitride with booming photocatalytic hydrogen evolution. Appl. Catal. B: Environ. 2019, 246, 61-71.

[50] Zhang, J., Huang, F. Enhanced visible light photocatalytic H2 production activity of g-C3N4 via carbon fiber. Appl. Surf. Sci. 2015, 358, 287-295.

[51] Zhang, L., Jin, Z., Lu, H., Lin, T., Ruan, S., Zhao, X. S., Zeng, Y.-J. Improving the Visible-Light Photocatalytic Activity of Graphitic Carbon Nitride by Carbon Black Doping. ACS Omega. 2018, 3, 15009-15017.

Comment 3: In real application, it is possible to collect enough quantity from based Porous for large scale application? Please discuss the avaialability and the porcessing costs. The authors should mention why they used this material.

Response 3: Thanks for your comment. In fact, one of the main reasons why we choose to use pollen as a precursor to prepare biochar photocatalysts is that the acquisition cost of pollen is low and its practical value is high. Moreover, there are many types of pollen and are ubiquitous in nature. So, we think it is possible to collect enough quantity from based porous for large scale application. And we chose lotus pollen for research, mainly because of its surface wrinkles and N, P self-doping phenomenon, which can improve the utilization of light. This has been pointed out in the paper to “Pollen is a biomass precursor with unique surface folds that can improve the utilization of light, and pollen carbon is rich in N and P elements, which can be retained in the pollen carbon matrix during the carbonization process.” (Lines 68-72, page 2)

Comment 4: The English should be checked carefully.

Response 4: Thanks for the valuable suggestion. We have scrutinized the paper for language expression and sentence structure issues and have made improvements to make our paper more readable.

Once again, we thank you and the reviewers for the time you put in reviewing our paper. We feel that your suggestions to be very valuable and have carefully revised them. Looking forward to our revised manuscript meeting your expectations.

Sincerely,

Can-Zhong Lu

Round 2

Reviewer 2 Report

The manuscript can be published in the present form.